# Effects of Safety Harnesses Protecting against Falls from a Height on the User’s Body in Suspension

**DOI:** 10.3390/ijerph20010071

**Published:** 2022-12-21

**Authors:** Krzysztof Baszczyński

**Affiliations:** Central Institute for Labour Protection—National Research Institute, Department of Personal Protective Equipment, Wierzbowa 48, 90-133 Łódź, Poland; krbas@ciop.lodz.pl

**Keywords:** falls from a height, full body harnesses, ergonomics, suspension trauma

## Abstract

The present work concerns the impact of safety harnesses on the human body in the context of suspension trauma. Phenomena at the man/harness interface were studied on a group of men professionally working at a height and using personal protective equipment (PPE). In the study, subjects wearing a safety harness were suspended for 3 min in controlled conditions. Three types of safety harnesses of different design were used. The harnesses were evaluated on the basis of the subjects’ opinions expressed in a questionnaire administered following trials. The most important phenomena observed were the compression exerted by textile straps, inconvenient body position, as well as straps tightening around the neck and torso. The results of trials involving human subjects were convergent and complementary with tests using an anthropomorphic dummy, enabling an evaluation of the basic designs of safety harnesses.

## 1. Introduction

Analyzing the diversity of work sites in the contemporary industry in terms of their spatial orientation, one can note that many of them are elevated above the surrounding level. This is mostly found in such sectors as the construction, energy, and mining industries, etc. Consequently, in many cases workers are at risk of falling from a height, which is consistent with data on workplace accidents [1]. For instance, according to the annual report of the Central Statistical Office for 2020 [2], in Poland there were a total of 4227 accidents involving workers falling from a height that year, of which 30 led to serious injury, and 31 to death. In the construction industry alone, there were 445 accidents of this kind, of which 11 were lethal and 11 serious, while in manufacturing there were 1040 accidents, of which 7 were lethal and 7 serious. The risk of falling from a height in industrial settings can be mitigated by a variety of methods. The most important of them include organizational-technical measures, collective protection measures, as well as personal protective equipment (PPE). Among these methods, PPE is of particular note due to its widespread use.

PPE kits against falling from a height always contain a harness to be worn by the user [3,4]. Depending on its design, a harness may be used to arrest a fall, prevent a fall from occurring, support the worker’s position, or enable work in a suspended position by rope access. In all of these cases, the user is supported by a harness, and so its elements, such as textile straps as well as adjustment and connecting buckles, compress the human body. In the case of fall-arrest harnesses complying with the standard EN 361:2002 [5], this compression occurs during the dynamic process of arresting the user’s fall and continues as the user remains suspended. In the case of work positioning and restraint harnesses meeting the requirements of EN 358:2018 [6], compression is mostly exerted under static loading on users supporting themselves by means of those harnesses. Finally, harnesses consistent with the standard EN 813:2008 [7] exert a static force on the user’s body in suspension.

The impact of a harness on the human body encompasses the pressure exerted by its constituent straps, by the forced positioning of the user suspension, as well as by attachment, connection, and adjustment elements (buckles) impacting the user during fall arrest. These phenomena may generate hazards to the user’s health, or even life. Of special note are hazards related to the user being suspended following a fall arrest, as he or she awaits assistance and evacuation from a height. These hazards are mostly associated with compression on the body surface and with mobility constraints, which may disrupt circulation, leading to loss of consciousness, or even death. These phenomena are related to suspension trauma, which is a state of shock caused by the passive suspension of a person in a safety harness [8,9]. This issue and articles devoted to it are presented in the next section of this article. Due to the gravity of the problem, the present study aims to evaluate the impact of commercially available safety harnesses on the user’s body in suspension.

## 2. State of the Art

Harness design largely depends on its intended purpose. The key structural features of harnesses are specified in the European standards on PPE protecting against falls from a height. The basic requirements and test methods for sit harnesses designed for the support of users performing rope access work are contained in the standard EN 813:2008 [7]. In addition to requirements concerning static and dynamic strength parameters, the standard also defines methods for testing ergonomic properties. Tests should also involve the evaluation of harness impact on the human body by human subjects wearing such harnesses. The basic requirements and test methods for work positioning and restraint harnesses are laid out in the standard EN 358:2018 [6]. In turn, special safety harnesses for mountaineering are described in EN 12277:2015+A1:2018 [10]. The most important type of harnesses designed for arresting falls in industrial settings are full body harnesses meeting the requirements of EN 361:2002 [5]. The design of safety harnesses protecting against falls from a height and issues associated with their use have been discussed in publications by Sulowski [3,4] and Baszczyński [11].

The application of PPE is one of the most widespread methods of protecting people against falling from a height. However, despite numerous advantages, this method also carries some specific hazards. Analysis of reports on accidents involving users donning PPE of this type reveals the problem of suspension trauma arising as a result of users being suspended in a harness; which has been discussed in publications [8,9,12,13,14]. This concerns both industrial settings as well as mountaineering and speleology. The problem occurs mostly in the case of using the harnesses specified in the standards EN 361:2002 [5], EN 813:2008 [7], and EN 12277:2015+A1:2018 [10]. Suspension trauma constitutes a serious hazard to human health and life, both following fall arrest and during purposeful suspension without any dynamic forces. This kind of trauma may lead to limb numbness, difficulty breathing, acute pain, loss of consciousness, and even death in the worst-case scenario. According to publications [14,15,16,17,18,19,20,21,22], suspension trauma predominantly affects individuals who pursue mountaineering and speleology. Studies presented in [23,24,25,26,27] indicate that the causes and severity of suspension trauma are associated both with the properties of the human body and external factors, such as harness design. The main factors associated with the user’s body cited in the aforementioned publications include:Anatomical features;Body dimensions and weight;Psycho-physical state;The influence of substances such as medications, alcohol, etc.;Loss of consciousness due to (e.g., impact against objects while falling from a height).

The most important external factors determining the occurrence and severity of suspension trauma are:
Suspension duration;Movement impairment during suspension;The angle between the user’s torso and the vertical;Leg position in suspension;Harness fit to the user’s body,Harness design, including harness attachment point position;Compression of the human body by harness straps and buckles.

The cited publications indicate that the most severe responses of the human body to suspension are attributable to the compression exerted by the constituent elements of harnesses, such as textile straps as well as adjustment and connecting buckles. The experimental results given in publications [28,29,30,31,32] indicate that anthropomorphic dummies are the most valuable tools in researching the mechanical phenomena associated with the impact of mechanical factors on the human body. The compression exerted by harnesses of different designs on the surface of the human body in suspension was studied using the anthropometric dummy Hybrid III 50M Pedestrian [33,34]. The pressure was measured by means of Tekscan devices [35], which enabled the mapping of pressure distribution on surfaces. The obtained results were presented in a paper by Baszczyński [36]. It was found that the greatest pressure was exerted by thigh straps in the crotch area of the dummy. It was also found that the main factors affecting the magnitude of the pressure were: safety harness design, its fit to the shape of the dummy, and the type of the attachment point (sternal or dorsal). A fall arrest study involving an anthropomorphic dummy wearing harnesses of different design is described in paper [37]. The study involved, inter alia, the measurement of the compression of the dummy surface under dynamic conditions using FujiFilm Prescale film [38], which changes its color tone depending on the contact pressure applied. The results showed a strong impact exerted by the textile straps used in the harness, and especially by their edges.

Analysis of current knowledge indicates that the examined phenomena are crucial from the standpoint of PPE users working at a height. Thus, the question arises as to whether the compression of the user’s body is the only adverse phenomenon in the case of suspension and how it is perceived by users (i.e., whether their experiences are convergent with the results of the tests conducted on anthropometric dummies). This paper presents findings from a study on the impact of harnesses protecting against falls from a height on the user’s body in suspension. The main objective of the study was to evaluate the effect of the basic designs of safety harnesses that are currently used in Poland on the experiences of their users in a state of controlled static loading.

## 3. Study Material

The study involved four typical designs of safety harnesses currently used in Poland for work at heights in a variety of workplaces. The harnesses are shown in Figure 1, and their basic design features are characterized in Table 1. The selected harnesses are primarily meant for fall arrest using dorsal (models H1, H2, H3, and H4) and sternal (models H2, H3, and H4) attachment points. Harness H4 can also be used for body positioning during work at a height due to lateral attachment points on its waist belt, as well as for work in a suspended position due to the presence of a ventral attachment point at the front of the waist belt, near the user’s center of gravity. In all of the models, the load-bearing (primary) straps are made from webbing from polyamide or polyester fibers with a width of at least 40 mm, with the auxiliary (secondary) straps having a width of at least 20 mm. In the case of harnesses H1, H2, and H3, their shoulder straps are continuous with the thigh straps, crossing at hip level. Additionally, these harnesses feature chest straps connecting the shoulder straps at sternum level. Harness H4 has thigh straps in the form of loops encircling the thighs, connected at the front and back with the shoulder straps.

All of the tested harnesses were of medium-large size and fit the human subjects.

## 4. Test Method

The effects of safety harnesses on a suspended human body were evaluated using the method specified in the standard EN 813:2008 [7]. In this method, a human subject wearing a safety harness is lifted by one of its attachment points, suspended for a set period of time, and then lowered and questioned about any adverse effects exerted by the harness on his or her body. Sample images of human subjects suspended in safety harnesses are presented in Figure 2.

According to the adopted method, all harness models and their attachment points were tested in the following steps:
The study participants were coached on the test procedure and safety precautions as well as told what they should pay attention to in terms of the effects of the harness on their body.The participants were familiarized with the harness; they donned it and the harness was fit to their body according to the manufacturer’s instructions.The participants performed several exercises such as forward bends, squats, and jumps to verify harness fit to their body.A 2 m long Kevlar rope was attached to the selected attachment point on the harness, with its other end being attached to the hook of a lifting device.After the participant’s consent, the lifting device was switched on and the participant was lifted approx. 10–15 cm above floor level.The participant remained suspended for 3 min (time was controlled with a stopwatch),During the test the participant and the harness were photographed.The participant was lowered to the floor and took off the safety harness.The participant filled out the questionnaire, thus recording the test results.The participant rested for approx. 30 min and performed light physical exercises to restore normal body function.

After the test, the participants filled out a questionnaire about their experience while being suspended in the safety harness; the questions concerned:
Points at which the straps, connecting buckles, attachment points, and other harness elements compressed the participant’s body (the participant was asked to mark those points on a chart);The degree of discomfort caused by the pressure of harness elements;Any uncomfortable body position forced by the harness;Any breathing difficulty;Numbness or tingling of body parts;Any hindrance to mobility in suspension;General comfort in suspension.

The degree of discomfort caused by the pressure of harness elements was assessed on a dedicated scale:

“1”—light pressure not causing discomfort;

“2”—pressure causing slight discomfort, acceptable over a period of more than 10 min;

“3”—pressure causing strong discomfort, bearable over a period of less than 10 min;

“4”—pressure causing strong discomfort involving limb numbness, difficulty breathing; etc.,

“5”—pressure causing pain as a result of which the participant had to be lowered to the floor less than 3 min after lifting.

Due to the potential health hazard to the participants, according to the guidelines contained in Annex A to the standard EN 813:2008 [7], all trials were conducted under the supervision of a doctor equipped with first aid medical equipment in case of the participant fainting or being hurt. The participant was elevated approximately 10–15 cm above the laboratory floor. The experimental station featured a 30 cm tall platform near the suspended participant so that he could stand up at any point in time. Furthermore, the supervising technician was ready to lower the participant immediately at any time in case of any danger.

Pursuant to the standard EN 813:2008 [7], the described harness tests require the participation of two human subjects differing in terms of weight by at least 30 kg and in height by at least 15 cm. Similarly, the study conducted at the BIA in Germany and described by Kloß [39] involved two subjects with three-axis accelerometers installed on their heads and near their center of gravity. In order to evaluate harnesses for a greater number of subject height and weight variants, the present study involved 10 men aged 30–59 who are professional firefighters specializing in height rescue involving the use of PPE against falling from a height. The participants reported that in their work they use different kinds of safety harnesses, both those protecting against falling from a height and those designed for work in a suspended position. The participants’ health as well as mental aptitude and physical fitness for work at a height were confirmed by appropriate medical certifications. The participants were 172–188 cm tall and weight from 72 to 100 kg, which is consistent with the requirements of the standard EN 813:2008 [7]. Each participant participated in eight harness suspension trials.

During the trials the participants wore light clothes that did not hinder their movements, such as T-shirts, track suit pants, and sports shoes.

## 5. Study Results

The questionnaires filled out by the participants were used to evaluate of the users’ experience while being suspended in different models of safety harnesses. The results consisted of the following elements:
Photographs of harnesses with points marked by the users as causing discomfort while in suspension;Description of the type of harness effect causing discomfort at a given point, such as thigh strap compression of the crotch area;L_w_ parameter representing the number of participants indicating a given point in the harness as causing noticeable discomfort, in relation to the total number of participants;W parameter representing the mean perceived degree of discomfort caused at a given point according to the scale presented in Section 4 of the article.

These results are presented in Figure 3, Figure 4, Figure 5 and Figure 6. Moreover, the participants’ observations from the trials are presented in Table 2.

Analysis of the test results indicates a number of recurrent observations made by the participants. The most important observations are presented below in descending order of frequency, from the most common to individual remarks.

Compression of the crotch area by the thigh straps. This phenomenon was mostly observed for suspension in harnesses H1, H2, and H3 using both the dorsal (X) and sternal (Y) attachment points. The edges of the thigh straps exerted pressure on the crotch area of the subject causing severe pain, which in one case led to quitting the trial before 3 min. The severity of pain caused by thigh strap compression of the crotch area was significantly lower in the case of harness H3, which featured cushioning pads. This phenomenon was the least pronounced in the case of harness H4, especially when suspended from the ventral attachment point, enabling the subject to adopt a sitting-like position. As a result, the subjects’ thighs remained vertical, while the thigh straps with cushioning pads were loaded across their width. Given literature reports, e.g., in publications [8,9,13,14,15,16,17,18,19,20,21,22,23,24,25,26,27], this phenomenon should be deemed one of the most dangerous in situations of humans being suspended in safety harnesses, as it may lead to serious circulatory disturbances in the legs. This was corroborated by leg numbness in one subject suspended in harness H1.Compression of the base of the neck by the shoulder straps. This phenomenon was mostly found for safety harnesses H1, H2, and H4 while using the dorsal (X) attachment point. It was caused by the fact that the shoulder straps were brought closer together at neck level as the dorsal attachment point shifted upwards when the user’s weight acted on the harness. This effect was not reported when sternal (Y) and ventral (Z) attachment points were used.Chest compression by the shoulder straps at clavicle level. This phenomenon was found for safety harnesses H1, H2, H3, and H4 while using the dorsal (X) attachment point, which forced the subject to lean forward. The pressure exerted by the shoulder straps on the chest in that position were perceived as very uncomfortable.Compression exerted by the sit strap connecting the thigh straps below the buttocks. This phenomenon was found for safety harnesses H2 and H3 while using the sternal (Y) and dorsal (X) attachment points. It was caused by the sit strap moving upwards as the user’s weight acted on the harness. As a result, the sit strap exerted pressure above the buttocks rather than support the user below them. However, compression from the sit strap was not experienced as very uncomfortable.Compression exerted by the shoulder and thigh straps crossing at hip level. This phenomenon was found for safety harnesses H2 and H3 while using the sternal (Y) attachment point. Forces acting on the shoulder and thigh straps brought them closer together, thus exerting pressure on the user’s body. Compression from the shoulder and thigh belts crossing at hip level was not perceived by the participants as very uncomfortable.Compression of the abdomen and inferior ribs by the waist belt (for positioning). This phenomenon was found for safety harness H4 while using the dorsal (X) attachment point. It was caused by the waist belt being pulled upward (as a result of the user’s body acting on the harness) and by the participant leaning forward due to the use of the dorsal attachment point.Posterior spinal hyperextension. This phenomenon was found for safety harness H3 while using the sternal (Y) attachment point. It was caused to the insufficient tensioning of the shoulder straps resulting in the upper part of the user’s back being unsupported. Thus, the participant had to correct his position by muscle contraction while suspended.

## 6. Summary

A summary of the presented results in terms of the relationship between safety harness design and the recorded observations of the study participants is given below:
Harness models H1, H2, and H3 in which the shoulder and thigh straps crossed at hip level caused compression of the crotch area while using both dorsal (X) and sternal (Y) attachment points. This phenomenon was also found in a study examining the pressure of thigh straps on an anthropometric dummy [36]. This is caused by the “vertical” orientation of the thigh straps and the resulting compression by the edges of textile straps. Both studies involving human subjects and an anthropometric dummy [36] showed that the use of cushioning pads (as in harness H3) alleviates the problem by reducing compression.Harness H4 generated the lowest degree of unacceptable thigh strap compression, especially when the participant was suspended from the ventral attachment point (Z). As that attachment point is situated near the human center of gravity, the user’s thighs were oriented horizontally, while thigh straps with cushioning pads were loaded across their entire width. This phenomenon was also observed in a study of thigh strap compression on the surface of an anthropometric dummy [36].In harness models H1, H2, and H4 suspended from the dorsal (X) attachment point, the edges of the shoulder straps exerted pressure on the base of the neck. This was caused by the shoulder straps coming too close together at neck level. From the standpoint of harness design, this resulted either from an inappropriate location of the dorsal attachment point (X) or its upward displacement due to harness loading.In some cases of suspension from the dorsal attachment point (X), the shoulder straps of harnesses H1-H4 compressed the front of the chest at clavicle level, which was also noted in paper [36]. In terms of safety harness design, this was attributable to the too low position of the dorsal attachment point (X) and the loosening of the shoulder straps.In harness models H2 and H3 compression was exerted by shoulder and thigh straps crossing at hip level when the sternal attachment point (Y) was used. This was caused by the forces acting on the shoulder and thigh straps, which brought them closer together, thus putting pressure on the user’s body. This effect was also reported from a study involving an anthropometric dummy [36].In harness H4, the waist belt (for work positioning) compressed the user’s abdomen and inferior ribs when suspended from the dorsal attachment point. This was caused by the upward movement of the waist belt in conjunction with the user leaning forward. From the standpoint of safety harness design, this phenomenon was attributable to excessive elasticity of the harness part (straps) below the waist belt, which was not able to prevent belt shifting under loading.Harness H3 suspended from the sternal attachment point (Y) caused the posterior spinal hyperextension of the suspended user. This was attributable to the insufficient tensioning of the shoulder straps and the low position of the attachment point, as a result of which the upper part of the user’s back remained unsupported. Analysis in terms of safety harness design indicates that the problem was caused by the fact that the frontal attachment point was located too close to the user’s center of gravity, as well as by the loosening of the shoulder straps (e.g., by the slippage of the textile straps in adjustment buckles).

## 7. Conclusions

Findings from the presented study involving human subjects are substantially convergent with those obtained from a study involving an anthropometric dummy [36]. These results show that harness design is essential to ensuring comfort and safety in suspension. The critical aspects of harness design in this respect are the arrangement of straps and their features such as width, the rigidity of their edges, and the presence of cushioning pads. From the standpoint of user comfort and safety in suspension, the best properties were found for harnesses with a ventral attachment point located near the user’s center of gravity. In contrast, the worst properties in this respect were exhibited by harnesses with dorsal attachment points.

The presented research results, in addition to scientific significance, also have practical applications since they can be used to design new constructions of safety harnesses that guarantee greater safety and comfort for their users.

There were several limitations in presented study. The first limitation concerned static test conditions. For safety reasons, the participants were lifted and lowered to the floor at low speed. This situation differs significantly from the real conditions of arresting a fall from a height and going into suspension of a user of a safety harness. The second limitation was related to the selection of safety harnesses for testing. Only harnesses intended for use in industrial environments, meeting the requirements of the standards EN 361:2002 [5] and EN813:2008 [7], were selected. The study was not extended to mountaineering harnesses. The third serious limitation was the use of only professional firefighters specializing in height rescue involving the use of PPE against falling from a height. In real working conditions in industry, employees using personal equipment protecting against falls from a height are often not as physically fit and trained as study participants, which may affect the results of the study.

In addition, during the tests, it was observed that the human subjects carefully fit the harnesses to their body. This means that the fit could significantly affect the comfort in the state of suspension. This problem has not been solved so far and it is planned to undertake research in this area. As a result of these studies, a measure of the fit of the harness to the user’s body and an assessment of its impact on comfort in the suspended state should be developed. Thanks to this, it will be possible to develop a procedure for checking the correct fit of the harness intended for users of personal equipment protecting against falls from a height. It is also planned to undertake study related to the assessment of the harness’s effect on the user’s body in dynamic conditions (i.e., during the fall arrest). These tests will take into account the participation of larger group of human subjects, various harness constructions as well as various types of connecting and shock-absorbing components affecting the fall arrest force [40,41,42].

## Figures and Tables

**Figure 1 ijerph-20-00071-f001:**
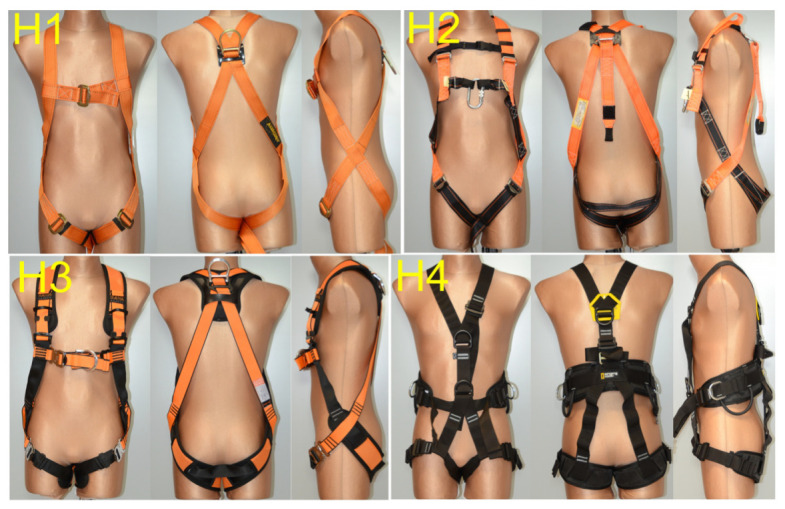
Safety harnesses tested in terms of their effects on the user’s body in suspension.

**Figure 2 ijerph-20-00071-f002:**
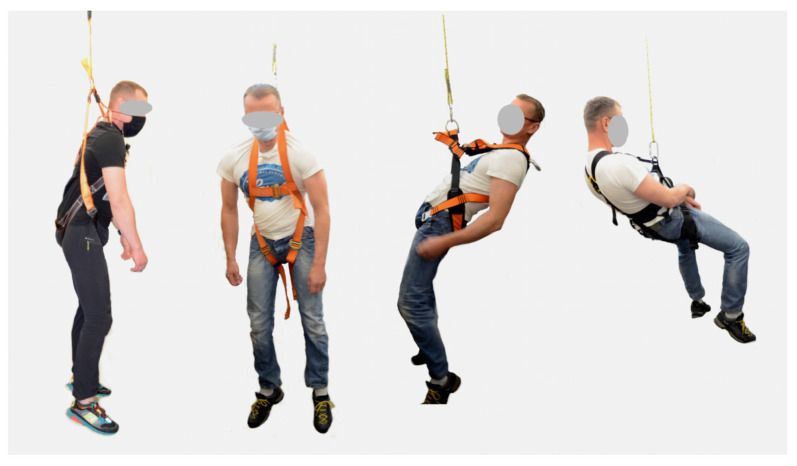
Sample images of human subjects suspended in safety harnesses.

**Figure 3 ijerph-20-00071-f003:**
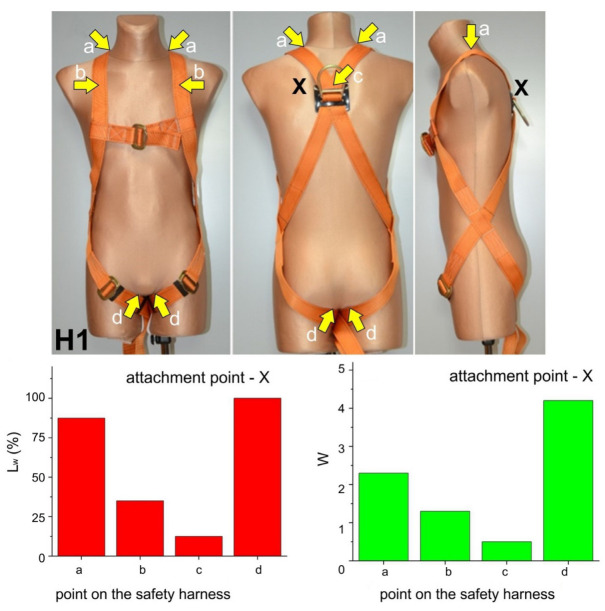
Test results for harness H1.

**Figure 4 ijerph-20-00071-f004:**
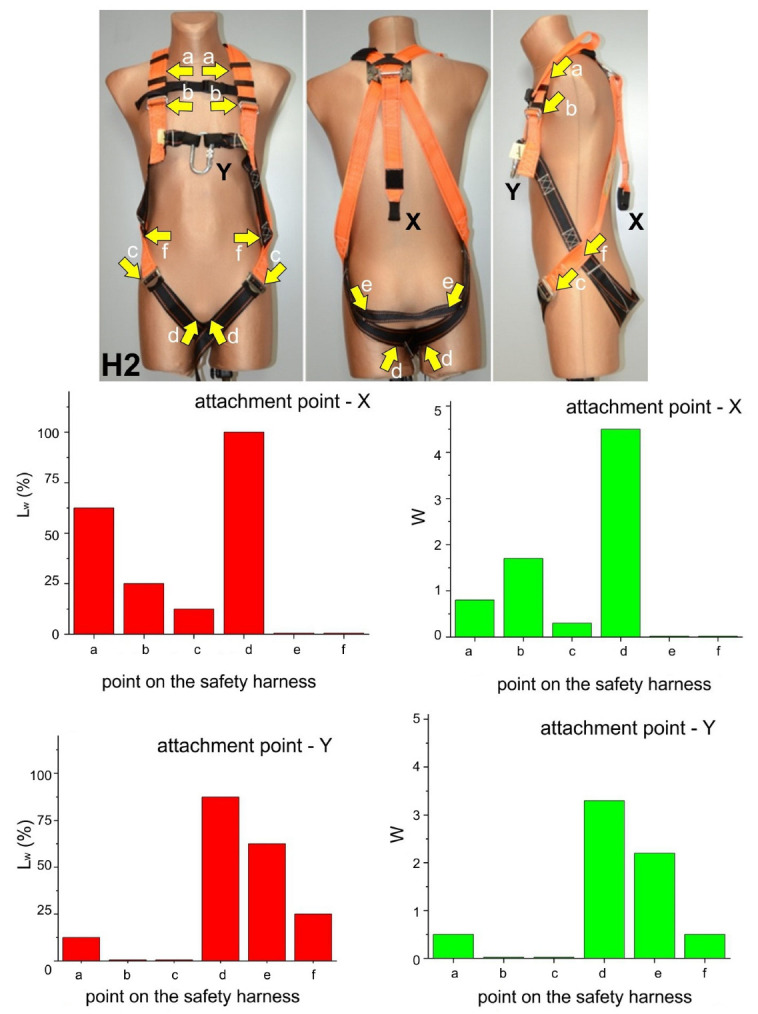
Test results for harness H2.

**Figure 5 ijerph-20-00071-f005:**
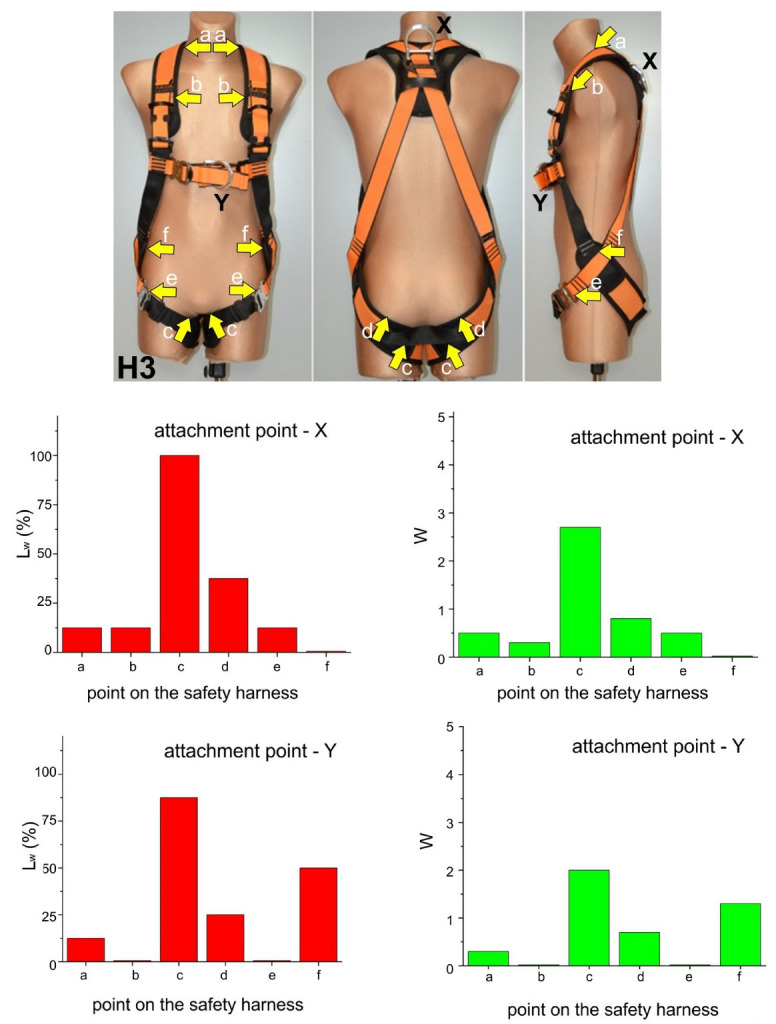
Test results for harness H3.

**Figure 6 ijerph-20-00071-f006:**
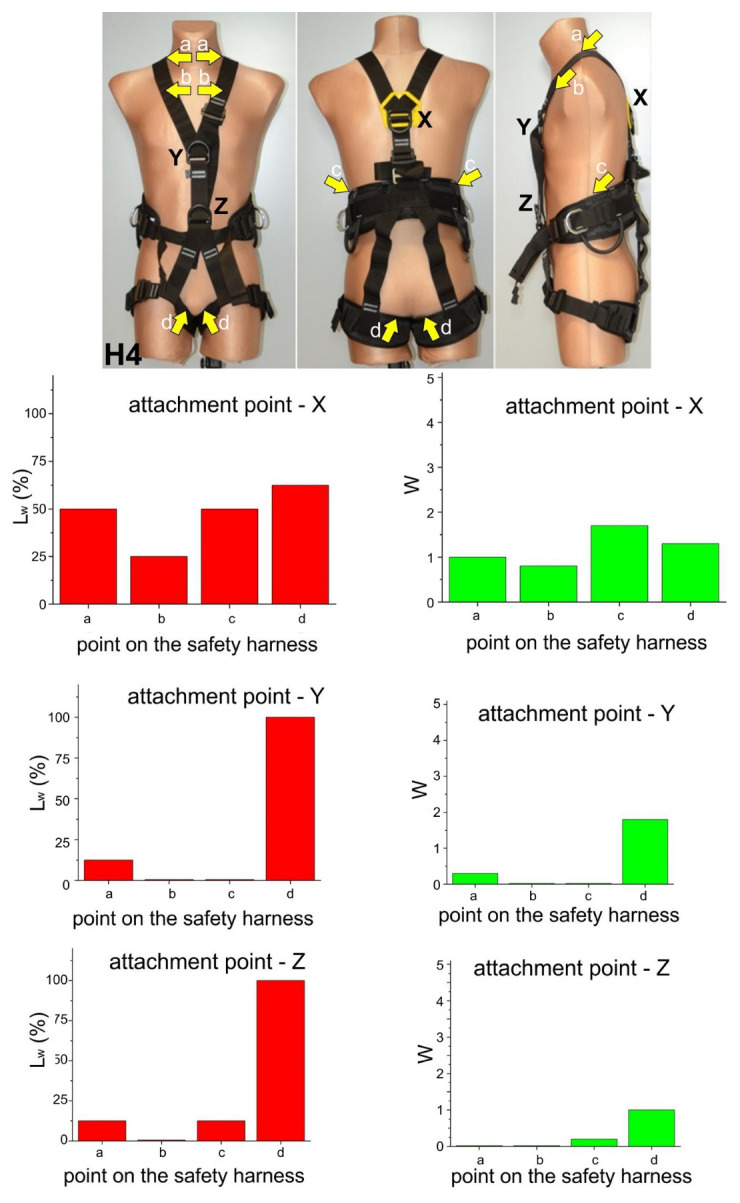
Test results for harness H4.

**Table 1 ijerph-20-00071-t001:** Safety harnesses used in the tests.

Symbol	Model	Manufacturer	Attachment Point			Cushioning Pads	Elements for Work Positioning	Other Design Features
			Dorsal	Sternal	Ventral			
H1	CA-101	Assecuro Sp. z o.o., Poland	+	−	−	−	−	Thigh straps crossing at the hip
H2	S-300	Lubawa S.A., Poland	+	+	−	−	−	Thigh straps crossing at the hip
H3	P451PO	Kaya Safety, Turkey	+	+	−	+	−	Thigh straps crossing at the hip
H4	Technic	Singing Rock s.r.o., Czech Republic	+	+	+	+	work positioning belt	Thigh straps in the form of closed loops

**Table 2 ijerph-20-00071-t002:** Participants’ observations from the trials.

Safety Harness	Attachment Point (Symbols as in Figure 3, Figure 4, Figure 5 and Figure 6)	Observation
H1	X	• Toe numbness (one case) • Chest compression by shoulder straps (one case)
H2	X	• Suspension time shortened to 2 min due to strong thigh pain caused by compression in the crotch area (one case)
	Y	• Sit strap shifted above the buttocks (four cases)
H3	X	• Shoulder straps tightening around the base of the neck (one case)
	Y	• Spinal pain in the lumbar area due to posterior hyperextension while suspended (three cases)
H4	X	• Upward shift of the waist strap causing compression of the abdomen and ribs (three cases)
	Z	• Downward shift of the posterior part of the thigh straps causing discomfort (one case)

## Data Availability

The data presented in this study are available on request from the corresponding author.

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
