# Peer review of "Effects of Safety Harnesses Protecting against Falls from a Height on the User’s Body in Suspension"

_ijerph, 2022, doi:10.3390/ijerph20010071_

Round 1

Reviewer 1 Report

I propose minor revisions

Author Response

Responses to reviewer's comments – Reviewer 1

  1. Comment: ”Introduction. I propose to include more information about “suspension trauma”, it is necessary that the is unconscious, motionless”

Response: The reviewer's comment was accepted and a brief note on suspension trauma was added in the "Introduction" section. It was pointed out that this issue was discussed in the chapter "State of Art" together with the citation of relevant publications.

  1. Comment: “State of Art. Some words about studies of the dynamin phenomena of the arrest with fall arrest systems must be included, we propose….”

Response: The reviewer's comment was accepted. In the Summary and Conclusions section, a fragment was added concerning the planned dynamic tests of the equipment protecting against falls from a height and references were made to 3 publications.

  1. Comment: “Test Method. Please explain why 3 minutes was the time selected”

Response: The test method for sit harnesses is specified in the EN813 standard. It requires suspending a person in the tested harness for no longer than 4 minutes. Since the properties of the harnesses used in the tests presented in the article were not known, the suspension time was shortened to 3 minutes for safety reasons.

  1. Comment: ”Need more details about what is Lw and W”.

Response: The reviewer's comment was accepted. Changes have been made to the descriptions of Lw and W parameters in Study Results section of the article.

  1. Comment: ” Graphics must be improved. Need a graphic result by attachment point, all harness at same graph, it will be easier to compare them”

Response: Unfortunately, it is not possible to present the test results of all harnesses in one graph. This is due to the fact that the results concern different points of contact between the harness and the human body, which were identified by the participants of the tests. Therefore, the results must remain separate for the individual harnesses and their attachment points.

  1. Comment: “Conclusions. Fitting has not been studied, non-conclusion can be done about fitting importance”

Response: The reviewer's comment was accepted and the " Summary and Conclusions" section was corrected.

  1. Comment: “Could be interesting compare prices and ergonomic results”

Response: I agree with the comment that the assessment of the relationship between the price of the harness and its ergonomic properties may be interesting from the point of view of users of personal equipment protecting against falls from a height. Unfortunately, the available data do not allow for such an analysis. Therefore, it is not possible to present this problem in the article.

Reviewer 2 Report

Some suggestions for future safety harnesses design based on this study should be highlighted in the conclusions.

Author Response

Responses to reviewer's comments – Reviewer 2

  1. Comment: “English language and style are fine/minor spell check required”

Response: The article was reviewed by a native English-speaking translator who has extensive experience in the revision of technical texts.

  1. Comment: “Some suggestions for future safety harnesses design based on this study should be highlighted in the conclusions”

Response: The reviewer's comment was accepted and in the section "Summary and Conclusions" information on the planned further work in the presented field was introduced.

Reviewer 3 Report

This study aimed to examine the human/wire harness interface phenomenon in a group of men who worked professionally at heights and used personal protective equipment (PPE), concerned with the impact of safety harnesses on the human body in the context of suspension trauma. The topic is interesting. Some comments for the authors to improve the quality of the manuscript are below.

1.      Add some references to support the cited examples and definite sentences, especially in the sessions of Introduction and State of the Art.

2.      Correct the format between words in the second paragraph of Introduction and the first paragraph in the State of the Art.

3.      it is better to begin sentences with capitalization in the sessions of State of the Art and Test Method.

4.      In the second paragraph of Test Method, please align the sentences.

5.      In the third paragraph of Test Method, please keep the opening indent, consistent with the rest of the paragraph format.

6.      In the fifth paragraph of Test Method, please correct the misspelling of words.

7.      In the session of Summary and Conclusions, please add the limitations of the experiments and future research directions.

Author Response

Responses to reviewer's comments – Reviewer 3

  1. Comment: “English language and style are fine/minor spell check required”

Response: The article was reviewed by a native English-speaking translator who has extensive experience in the revision of technical texts.

  1. Comment: “Add some references to support the cited examples and definite sentences, especially in the sessions of Introduction and State of the Art”

Response: The reviewer's comment was accepted and the sections Introduction and State of the Art The were supplemented with an appropriate references.

  1. Comment: “Correct the format between words in the second paragraph of Introduction and the first paragraph in the State of the Art”

Response: The reviewer's comment was accepted.

  1. Comment: “It is better to begin sentences with capitalization in the sessions of State of the Art and Test Method”

Response: The reviewer's comment was accepted.

  1. Comment: “In the second paragraph of Test Method, please align the sentences”

Response: The reviewer's comment was accepted.

  1. Comment: “In the third paragraph of Test Method, please keep the opening indent, consistent with the rest of the paragraph format”

Response: The reviewer's comment was accepted.

  1. Comment: “In the fifth paragraph of Test Method, please correct the misspelling of words”

Response: The reviewer's comment was accepted.

  1. Comment: “In the session of Summary and Conclusions, please add the limitations of the experiments and future research directions”

Response: The reviewer's comment was accepted and the " Summary and Conclusions" section was supplemented with information on limitations and research that will be conducted in the future.